# The role of personality traits on self-medicated cannabis in rheumatoid arthritis patients: A multivariable analysis

**José R. Galindo-Donaire**[1,2], **Gabriela Hernández-Molina**[1], **Ana Fresán Orellana**[3], **Irazú Contreras-Yáñez**[1], **Guillermo Guaracha-Basáñez**[1,2,4], **Oswaldo Briseño-González**[5], **Virginia Pascual-Ramos**[1]*

1 Department of Immunology and Rheumatology, Instituto Nacional de Ciencias Médicas y Nutrición "Salvador Zubirán", Mexico City, Mexico, 2 Postgraduate Studies Division, Medicine Faculty, Universidad Nacional Autónoma de México, Mexico City, Mexico, 3 Subdirectory of Clinical Investigations, Instituto Nacional de Psiquiatría "Ramón de la Fuente Muñiz", Mexico City, Mexico, 4 Emergency Medicine Department, Instituto Nacional de Ciencias Médicas y Nutrición "Salvador Zubirán", Mexico City, Mexico, 5 Department of Quality and Education, Psychiatric Hospital "Dr. José Torres Orozco", Morelia, Mexico

☯ These authors contributed equally to this work.

* virtichu@gmail.com

**Data Availability Statement:** All relevant data are within the manuscript and its Supporting Information files.

## Abstract

### Background

Rheumatoid arthritis (RA) patients commonly report medicinal cannabis use (MCU). Personality has been independently associated with both RA-related outcomes and MCU, but there is no information available on how they interact in RA patients. This study aimed to investigate a potential association between personality traits and MCU in RA outpatients, as well as to identify additional factors associated with its use.

### Methods

This cross-sectional study was performed between June 2020 and August 2021. Consecutive RA outpatients had standardized evaluations using an interview format to collect socio-demographic information, comorbidities, risk of recreational substance use, RA-related disease activity/severity, health-related quality of life, depressive and anxiety symptoms, five personality traits, and MCU in the 12 months before the interview. Multivariable logistic regression estimated adjusted odds ratios (aOR). The study was IRB-approved.

### Results

180 patients were included; 160 (88.9%) were women with a mean age of 53.4 ± 13 years. Fifty-three (29.4%) patients reported MCU. Among them, 52 (98.1%) used topical formulations. Neuroticism had the highest overall score ($\bar{x} = 3.47 \pm 0.34$). Openness to experience trait was higher in MCU patients in the comparative analysis (p = 0.007). In the multivariable regression, higher openness trait (aOR: 2.81, 95%CI: 1.11–7.10) along with moderate risk in tobacco use (aOR: 3.36, 95%CI: 1.04–10.7) and higher RA disease activity/severity (aOR: 1.10, 95%CI: 1.01–1.19) were independently associated with MCU.

**Funding:** The author(s) received no specific funding for this work.

**Competing interests:** The authors have declared that no competing interests exist.

## Conclusions

In the current study, personality influenced the seeking of MCU for pain relief, associating dynamically with higher disease activity/severity and tobacco use. Contrary to other available information, it did not relate to psychopathology or the recreational use of cannabis. Proactive interdisciplinary clinical evaluations around MCU in RA outpatients should include personality, besides standard clinical assessments, to understand patients' motivations for its use as they may reveal important clinical information.

## Background

Rheumatoid arthritis (RA) is a chronic systemic inflammatory disorder with articular and extra-articular involvement that can lead to significant joint structural damage, disability, reduced quality of life (QoL), and increased mortality [1, 2]. Current treatment recommendations highlight adherence to prescribed regimens with disease-modifying anti-rheumatic drugs (DMARDs) intended to achieve adequate disease activity control and prevent damage [1, 3]. However, patients also report using complementary and alternative medicines (CAM), including medicinal cannabis use (MCU), to treat RA [3]. Some cited reasons for alternative use have been the chronic and painful nature of the disease, adverse events associated with DMARDs use, and the scientific perception that the disease lacks a known cure [3–5], in spite of recent evidence that sustained drug-free remission, a proxy for a cure, can be achieved in particular patients [6]. Some RA patients might even use cannabis instead of their prescribed therapeutical regimens [7], while non-adherence to prescribed medication negatively impacts disease course and outcomes [8, 9].

Personality builds upon biological and environmental factors and encompasses a complex pattern of cognitions, emotions, and behaviors that characterize an individual [10]. The Big Five personality theory is a well-known hierarchically organized taxonomy to describe and assess personality, which derives from the lexical hypothesis [11] and has a long trajectory in psychological trait theory. According to this model, personality is conceptualized under five high-order domains (also referred to as factors or traits) that do not focus on psychopathology, but rather represent personality at the broadest level of abstraction [12]. Each domain includes many distinct and more specific personality attributes called facets [11]. These five domains (or traits) are extraversion, agreeableness, conscientiousness, neuroticism, and openness to experience. Extraversion summarizes facets related to activity and energy, dominance, sociability, expressiveness, and positive emotions. Agreeableness contrasts prosocial orientation with antagonism and includes facets such as altruism, tendermindedness, trust, and modesty. Conscientiousness describes socially determined impulse control that facilitates task- and goal-directed behavior. Neuroticism contrasts emotional stability with a broad range of negative affect, including anxiety, sadness, irritability, and nervous tension. Finally, openness to experience describes the complexity of an individual's mental and experiential life, intellect, culture and usually refers to being reflective, curious, perceptive, thoughtful, and analytical [12, 13].

Personality influences important life outcomes and chronic illness-related outcomes [14]. In the clinical context of RA, personality is important in terms of illness-perception [15], treatment acceptance [16], medication adherence [17], quality of life (QoL) [17], and self-care [18]. On the other hand, it has also been associated with cannabis use [19]. Neuroticism, extraversion, and openness traits have positively correlated with cannabis consumption, while

agreeableness and conscientiousness have negatively correlated [20, 21]. These findings have been primarily described for the recreational use of cannabis, yet it is important to recognize that complex situations arise regarding conduct, motivations, and self-reported intentions for its use. Concurrent medical and non-medical use is possible [22]. Also, cannabis relates to mental illnesses such as psychotic spectrum disorders [23], substance use disorders [24], and mood disorders [25].

With the above considerations in mind, this study's primary objective was to investigate a potential association between personality traits and MCU in RA outpatients. We hypothesized that higher scores in extraversion, neuroticism, and openness traits as well as lower scores in agreeableness and conscientiousness would be associated with MCU in these patients. Also, additional demographic and clinical associated factors would be explored.

## Methods

### Participants

This cross-sectional study was performed between June 2020 and August 2021 at the outpatient clinic of the Department of Immunology and Rheumatology (OCDIR) of the *Instituto Nacional de Ciencias Médicas y Nutrición "Salvador Zubirán"* (INCMNSZ), a tertiary care national referral center for rheumatic diseases, in Mexico City.

Consecutive RA patients attending the OCDIR were invited to participate. RA diagnosis was determined as per the electronic health record (EHR), based on the treating rheumatologist's criteria. Exclusion criteria were an overlapping rheumatologic syndrome (except Secondary Sjögren Syndrome), current treatment for any cancer (palliative care included), presence of any substance use disorder (SUD) or recreational cannabis use in the past 12 months, ever history of severe mental illness (that included psychotic manifestations) or current severe psychiatric diagnosis that hindered interview performance or completion.

### Measures and procedures

All outpatients who consented to participate were invited to a psychiatric interview to collect sociodemographic information, relevant psychiatric and comorbid conditions, disease and treatment-related information, and legal/illegal substance use/abuse. We considered personality as the exposure variable and used the Big Five Inventory (BFI) to assess personality traits [26]. We evaluated recent (last 12 months) MCU as the primary outcome and used a locally adapted version of the International Complementary and Alternative Medicine Questionnaire (ICAM-Q) to assess it [5]. Additional information was gathered in the interview through cross-culturally validated instruments to account for risk in recreational substance use (ASSIST) [27], RA-disease activity/severity (RAPID-3) [28], QoL related to RA (QoL-RA) [29], depressive (PHQ-9) [30] and anxiety (GAD-7) [31] symptoms, current psychiatric illness severity (CGI-S) [32], and comorbid medical conditions (Charlson comorbidity index) [33]. EHR was retrospectively reviewed considering a time frame of 12 months before the interview to evaluate the presence of current treatment and follow-up for any psychiatric diagnosis as well as to additionally complement the data provided by the patients regarding medication use and chronic comorbidities. A summary description of instruments is presented in the **Supplementary data in S1 Appendix**.

All interviews were performed by a single Consultation-Liaison psychiatrist (JRGD) within a two-week interval since patients' last rheumatology visit. The interviewer was not involved in patient care. Due to the COVID-19 pandemic, participants decided on an in-person interview or a videoconference interview.

## Groups and definitions

MCU was defined based on responses in the Spanish version of the (modified) ICAM-Q. The Spanish version has four main sections: (1) recent visits to different providers of Complementary and Alternative (CAM) treatment; (2) CAM treatment received from a physician; (3) consumption of medicinal products derived from herbs, vitamins, minerals, or homeopathic medicines; and (4) self-help practices implemented by the patients. For the current study, we used only section 3 and directed patients' responses to MCU exclusively. In section 3, patients were asked about the use of any product containing cannabis or derivatives within the past 12 months. Then, patients were also asked to identify the main reason for MCU and to evaluate how helpful they found the product. Medical use was defined when patients selected the "symptoms-related use" option.

Finally, MCU was defined when patients responded positively in section 3, for any route of administration of medicinally motivated cannabis products within the 12-month time frame before the interview. Our comparison groups (MCU, Non-MCU) were established according to these responses.

## Sample size calculation

To detect an effect size of at least 20% for the absolute difference in MCU (dependent variable) between high or low levels for any personality trait (independent variable), we estimated the sample size using a two-tailed test, a 5% significance level, and a power of 80%. The expected proportions for MCU between high vs. low-level personality traits, according to the mean score ± one standard deviation (SD) using within sample local norms, were 31% and 11%, respectively, based on data from a nationwide survey where 21% of arthritis patients described MCU [4]. The G*Power estimate was a minimum sample size of at least 142 patients, which was increased to 170 patients accounting for 20% of potentially non-analyzable data.

Additionally, the sample size was estimated, considering personality traits as a continuous variable. We ought to detect an effect size of 0.5, as a medium effect size convention [34], in the absolute difference in median scores for each personality trait between patients with and without MCU. The sample size was estimated using a two-tailed test, an allocation ratio n2/n1 of 1.57, a significance level of 5%, and a power of 80%. The G*Power estimate was a sample size of 136 patients.

With both exercises, the final sample obtained and patient distribution allowed us to have a ≥95% power for a two-tailed test.

## Statistical analysis

Frequencies and percentages are presented for categorical variables and central tendencies with dispersion for continuous variables in the descriptive analysis. Student's t-test or Mann-Whitney's U compared normally or non-normally distributed continuous variables (according to Shapiro Wilk's test). Fisher's exact or $X^2$ test was used when comparing categorical variables.

A comparative analysis of all variables was performed between cannabis users and non-users. Statistically significant differences in the comparative analysis and clinical relevance determined variable inclusion in the multivariable logistic regression. For this multivariable analysis, we initially conceived the global model (model A) with all variables. Then, a test-based backward selection procedure was used to define two additional models, derived from the global model. For this selection, Wald´s test value (Z-score), which indicates the magnitude and precision of each variable´s coefficient estimate and measures their effect on the response variable (MCU), was used as criteria for exclusion/elimination in the backward selection

procedure. Current psychiatric diagnosis was retained as a covariable across all models, this was based on clinical reasoning due to its frequent association with cannabis use. The procedure began with Model A which included tobacco use, RAPID-3, QoL-RA, PHQ-9, and GAD-7 scores, current psychiatric diagnosis, the five personality traits scores, and psychiatric interview modality. Then, variables with a bigger effect (Z-scores > ±2) in model A were excluded to define a second model (model B). In contrast, a third model (model C) excluded variables with a smaller effect (Z-scores < ±1). Raw scores were preferred for each continuous variable included. Akaike's Information Criterion (AIC) was used to compare and determine the best model, avoiding overfitting or underfitting [35]. McFadden's pseudo-$R^2$ and Hosmer-Lemeshow test are reported as additional measures of goodness of fit and C-statistic (AUC) for the overall accuracy.

There were no missing data. Statistical tests were two-sided and evaluated at a 0.05 significance level. Results for multivariable regression are expressed as adjusted odds ratios (aORs), which are exponentiated regression coefficients (expβ), and their 95% confidence interval. All analyses were performed using STATA (version 16.0, Stata Corp LLC, College Station, TX) and SPSS (version 21.0, IBM Corp., Armonk, NY, USA).

### Ethics

The study was conducted according to the principles of the Declaration of Helsinki. The Research Ethics Committee of the INCMyN-SZ approved the study (reference number: IRE-3319). Written informed consent was obtained from all the patients who had an in-person interview. Due to the COVID-19 pandemic, for those patients who were interviewed by videoconference the committee waived written consent and verbal authorization was requested to participate. Patients were able to withdraw consent at any stage of the study.

## Results

### Population characteristics

There were 240 RA outpatients invited to participate and 183 (76.2%) consented. Three of them scored CGI-S ≥6 and were excluded. Finally, 180 patients were analyzed among whom 70 (38.9%) had an in-person interview and 110 (61.1%) opted for a videoconference interview (shown in **Fig 1**).

**Table 1** summarizes the population's sociodemographic and clinical characteristics. Participants were primarily middle-aged women, living with a partner, with economic dependency, a long-standing disease, with comorbid conditions, and median RAPID-3 and QoL-RA total scores of 9.3 and 7.8 due to non-normal distribution, respectively. Spearman's correlation (Rho, ρ) of RAPID-3 total score was high with its pain subscale (ρ = 0.93, P<0.001), self-reported joint count subscale (ρ = 0.85, P<0.001), and patient global estimate subscale (ρ = 0.94, P<0.001). Most patients were on DMARDs, while one-third were on corticosteroids. Regarding mental health, one in ten patients had a current psychiatric diagnosis (the totality of them had CGI-S score≤5), 16 (8.9%) patients were in the moderate risk category for tobacco use, and median PHQ-9 and GAD-7 scores translated into an absence of significant symptoms for most participants.

Because interview modalities differed, patients' characteristics were compared between those who opted for an in-person or a videoconference interview. Overall, patients from both groups were similar except for age, education, comorbidity, and the openness trait (**Table 2**). Interview modality was included in the multivariable analysis.

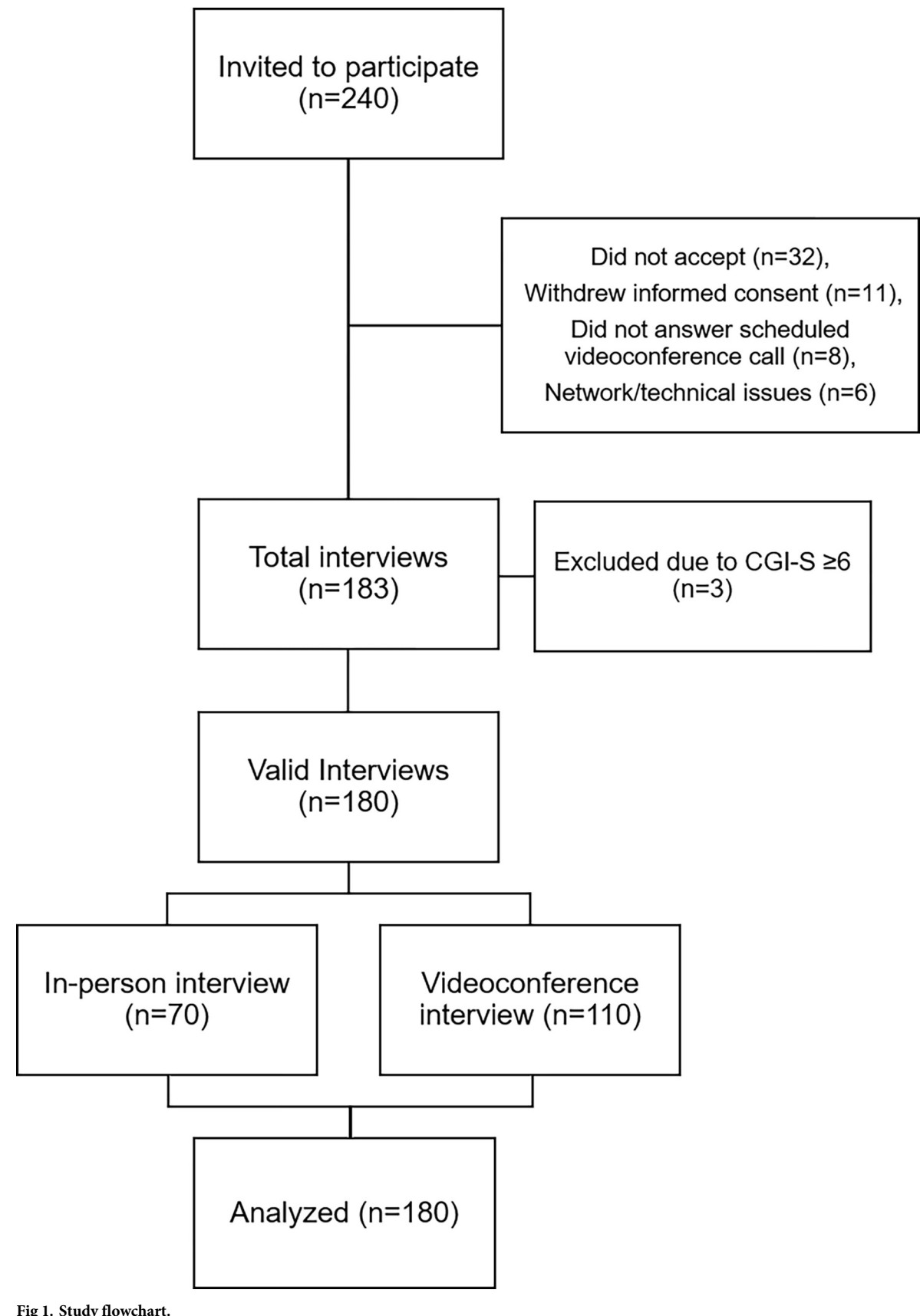

**Fig 1. Study flowchart.**

**Table 1. Population characteristics and comparison between patients with and without medicinal cannabis use (MCU).**

| Characteristics | Overall population (n = 180) | Patients without MCU (n = 127) | Patients with MCU (n = 53) | P |
|---|---|---|---|---|
| **Sociodemographic** | | | | |
| Years of age* | 53.4 ± 13 | 53 ± 12.7 | 54.5 ± 13.8 | 0.48 |
| Female sex | 160 (88.9) | 111 (87.4) | 49 (92.4) | 0.32 |
| Years of formal education* | 11.7 ± 4.1 | 11.8 ± 4.3 | 11.3 ± 3.7 | 0.50 |
| Living with a partner | 83 (46.1) | 54 (42.5) | 29 (54.7) | 0.13 |
| Economic dependency | 78 (43.3) | 52 (40.9) | 26 (49.1) | 0.31 |
| **RA-related** | | | | |
| Years of disease duration† | 15 (9–24) | 14 (8–25) | 15 (10–21) | 0.58 |
| Years of follow-up at the OCDIR† | 11 (7–18.5) | 10 (6–20) | 12 (8–18) | 0.26 |
| ≥1 comorbidity | 128 (71.1) | 90 (70.9) | 38 (71.7) | 0.91 |
| Charlson score† | 0 (0–0) | 0 (0–0) | 0 (0–0.5) | 0.68 |
| Corticosteroids use | 70 (38.9) | 55 (43.3) | 15 (28.3) | 0.06 |
| DMARDs use | 175 (97.2) | 122 (96.1) | 53 (100) | 0.32 |
| N° DMARDs/patient† | 1 (1–2) | 1 (1–2) | 2 (1–2) | 0.16 |
| Recent DMARD increase | 55 (30.6) | 33 (26) | 22 (41.5) | 0.09 |
| RAPID-3† | 9.3 (3–13.7) | 7.3 (1.5–13.3) | 10.7 (7.3–14.3) | 0.004 |
| QoL-RA† | 7.81 (6.4–9.1) | 8.12 (6.3–9.2) | 7.5 (6.5–8.3) | 0.04 |
| Previous hospitalizations | 13 (7.2) | 9 (7.1) | 4 (7.5) | 1.00 |
| Satisfaction with medical care | 166 (92.2) | 118 (92.9) | 48 (90.5) | 0.55 |
| **Mental health-related** | | | | |
| ASSIST (moderate risk category) | | | | |
| Tobacco use | 16 (8.9) | 7 (5.5) | 9 (16.9) | 0.02 |
| Alcohol use | 7 (3.9) | 4 (3.1) | 3 (5.6) | 0.42 |
| Other substance use | 1 (0.6) | 1 (0.8) | 0 (0) | 1.00 |
| Any current psychiatric diagnosis | 22 (12.2) | 11 (8.6) | 11 (20.7) | 0.02 |
| PHQ-9 score† | 4 (1–8) | 4 (1–7) | 6 (2–12) | 0.02 |
| GAD-7 score† | 3 (1–5.5) | 2 (0–5) | 4 (2–6) | 0.02 |
| Extraversion trait score* | 3.01 ± 0.3 | 3 ± 0.30 | 3.02 ± 0.32 | 0.74 |
| Agreeableness trait score* | 3.16 ± 0.38 | 3.14 ± 0.37 | 3.19 ± 0.39 | 0.43 |
| Conscientiousness trait score* | 2.93 ± 0.41 | 2.95 ± 0.37 | 2.89 ± 0.50 | 0.35 |
| Neuroticism trait score* | 3.47 ± 0.34 | 3.47 ± 0.33 | 3.48 ± 0.35 | 0.82 |
| Openness trait score* | 3.03 ± 0.42 | 2.97 ± 0.40 | 3.16 ± 0.44 | 0.007 |
| Video conference psychiatric interview modality | 110 (61.1) | 80 (62.9) | 30 (56.6) | 0.42 |

Data are presented as n (%) unless specified otherwise.

*Mean ± SD

†Median (IQR) MCU = Medicinal cannabis use; OCDIR = Outpatient clinic of the Department of Immunology and Rheumatology; DMARD = Disease modifying anti-rheumatic drugs; RAPID-3 = Routine Assessment of Patient Index Score-3; QoL-RA = Quality of Life in RA scale; ASSIST = Alcohol, smoking and substance involvement screening test; PHQ-9 = Patient Health Questionnaire 9; GAD-7 = Generalized Anxiety Disorder.

## MCU and related information

There were 53 (29.4%) patients who reported MCU, and the different administration routes were not exclusive and could be overlapped. Among them, 52 (98.1%) reported topical MCU, nine (17%) took it orally, and three (5.7%) smoked it. Among medicinal cannabis users, symptoms-related to the underlying rheumatic disease were the primary motivation in 47 (88.7%) patients.

**Table 2. Comparison of characteristics between patients with in-person and videoconference interview.**

| Characteristics | Patients with in-person interview (n = 70) | Patients with videoconference interview (n = 110) | P |
|---|---|---|---|
| **Sociodemographic** | | | |
| Years of age* | 55.7 ± 13.0 | 51.9 ± 12.8 | 0.05 |
| Female sex | 63 (90) | 97 (88.0) | 0.70 |
| Years of formal education* | 10.9 ± 4.3 | 12.2 ± 4 | 0.04 |
| Living with a partner | 33 (47.1) | 60 (45.5) | 0.82 |
| Economic dependency | 30 (42.9) | 48 (43.6) | 0.91 |
| COVID-19-related change in economic activities | 29 (34.3) | 26 (23.6) | 0.12 |
| **RA-related characteristics** | | | |
| Years of disease duration† | 14.5 (9–24) | 15 (9–24) | 0.85 |
| Years of follow-up at the OCDIR† | 12 (8–20) | 10 (6–18) | 0.33 |
| Charlson score† | 0 (0–1) | 0 (0–0) | 0.003 |
| ≥1 comorbidity | 56 (80) | 72 (65.5) | 0.03 |
| Corticosteroids use | 31 (44.3) | 39 (35.5) | 0.23 |
| DMARDs use | 67 (95.7) | 108 (98.2) | 0.37 |
| N° DMARDs/patient† | 1 (1–2) | 2 (1–2) | 0.36 |
| Recent DMARD increase | 22 (31.4) | 33 (30) | 0.30 |
| RAPID-3† | 9.5 (2–14) | 8.85 (3.5–13.7) | 0.95 |
| QoL-RA† | 8 (6–9.3) | 7.75 (6.5–8.9) | 0.99 |
| Previous hospitalizations | 5 (7.1) | 8 (7.3) | 0.97 |
| Satisfaction with medical care | 67 (95.7) | 99 (90.0) | 0.16 |
| **Mental health-related** | | | |
| ASSIST (moderate risk category) | | | |
| Tobacco use | 3 (4.3) | 13 (11.8) | 0.08 |
| Alcohol use | 4 (5.7) | 3 (2.7) | 0.43 |
| Other substance use | 0 (0.0) | 1 (0.9) | 1.00 |
| Current psychiatric diagnosis | 7 (10) | 15 (13.6) | 0.46 |
| PHQ-9 score† | 5 (1–8) | 4 (1–8) | 0.55 |
| GAD-7 score† | 2 (1–5) | 3 (1–6) | 0.43 |
| Extraversion trait score* | 2.98 ± 0.30 | 3.02 ± 0.3 | 0.63 |
| Agreeableness trait score* | 3.20 ± 0.34 | 3.13 ± 0.40 | 0.20 |
| Conscientiousness trait score* | 2.97 ± 0.44 | 2.91 ± 0.39 | 0.38 |
| Neuroticism trait score* | 3.43 ± 0.34 | 3.49 ± 0.33 | 0.26 |
| Openness trait score* | 2.89 ± 0.39 | 3.12 ± 0.41 | <0.001 |
| MCU | 23 (32.9) | 30 (27.3) | 0.42 |
| MCU disclosure with rheumatologist | 6/23 (26.1) | 4/30 (13.3) | 0.30 |

Data are presented as n (%) unless specified otherwise.

*Mean ± SD

†Median (IQR). MCU = Medicinal cannabis use. OCDIR = Outpatient clinic of the Department of Immunology and Rheumatology. DMARD = Disease modifying anti-rheumatic drugs. RAPID-3 = Routine Assessment of Patient Index Score-3. QoL-RA = Quality of Life in RA scale. ASSIST = Alcohol, Smoking and Substance Involvement Screening Test. PHQ-9 = Patient Health Questionnaire 9. GAD-7 = Generalized Anxiety Disorder.

MCU was combined with institutional health care. Ten (18.9%) patients were already users before RA diagnosis, while the rest became users after it. MCU was also restricted to RA-related acute pain crises (n = 46, 86.8%) and patients perceived at least some related benefit (n = 43, 81.1%). Moreover, three (5.7%) patients mentioned ever tapering rheumatologic treatment. From the total sample, 111 (61.7%) agreed that general MCU should be disclosed with

the primary rheumatologist. This declined among MCU patients (n = 23, 43.4%) and ultimately only 10 (18.9%) of them had communicated its use to their primary physician.

## Personality traits in the target population

**Table 1** summarizes the personality trait scores of the sample. The neuroticism trait had the highest score while the conscientiousness trait was the lowest. Cut-off points based on local norms classified 30 (16.7%) patients with high-level extraversion, 32 (17.8%) with high-level agreeableness, 25 (13.9%) with high-level conscientiousness and neuroticism, and 30 patients (16.7%) with high-level openness.

## Factors associated with MCU

MCU patients' characteristics were compared to those without MCU. Results are detailed in **Table 1**. In the former group, they had higher disease activity/severity and reduced QoL. They used less frequently corticosteroids but had a higher proportion of recent DMARD dose increases. Health care satisfaction was high among MCU patients and did not differ with non-users. Current psychiatric diagnosis, PHQ-9, GAD-7, and the openness to experience trait scores were higher in patients with MCU. More frequent tobacco use was observed in MCU patients (**Table 1**). No other proportion in drug use, including alcohol, was different. Unadjusted ORs for personality scores on MCU are presented in **Fig 2**.

## Multivariable analysis

**Table 3** summarizes the aORs from multivariable logistic regression. Model C exhibited the best model fit (AIC = 209.5) and postestimation statistics indicated an overall good model. Area under the curve for this model was 0.71 as shown in Table 3 and **Fig 3**. For this model, moderate risk in tobacco use (P = 0.04), openness personality trait score (P = 0.02), and RAPID-3 score (P = 0.02) were independently associated with MCU. A similar pattern was observed in model A where the openness trait showed a statistical trend (aOR = 2.46, 0.95–6.35, P = 0.06).

## Discussion

This study was conducted on patients from a tertiary care referral center in Mexico City, which might be considered to represent RA outpatients. Assessments had no interobserver variability and were standardized to enable a comprehensive and unbiased approach to rheumatic disease, mental health-related outcomes, and MCU under real-life conditions. The study found that recent MCU was frequent, in almost one-third of our patients. Neuroticism had the highest overall scores but only higher openness to experience personality trait scores were consistently associated with MCU. Additional factors included higher RA disease activity/severity and moderate risk in tobacco use.

Regarding MCU, the observed percentage of use is similar to other findings across different populations with rheumatic diseases [4, 36, 37], although lower percentages have also been reported in Canadian patients [38, 39]. As for Mexico, there are no published contributions on MCU specifically, but there is existing literature on CAM in rheumatic diseases where topical marijuana is mentioned in 8–31% of patients [40, 41]. Interestingly, our patients preferred topical MCU where no psychoactive properties are expected. This is in contrast with a scoping review that addressed users' characteristics and their perceptions about cannabis which found inhalation as the most frequent pattern for chronic musculoskeletal pain [37]. Also, our

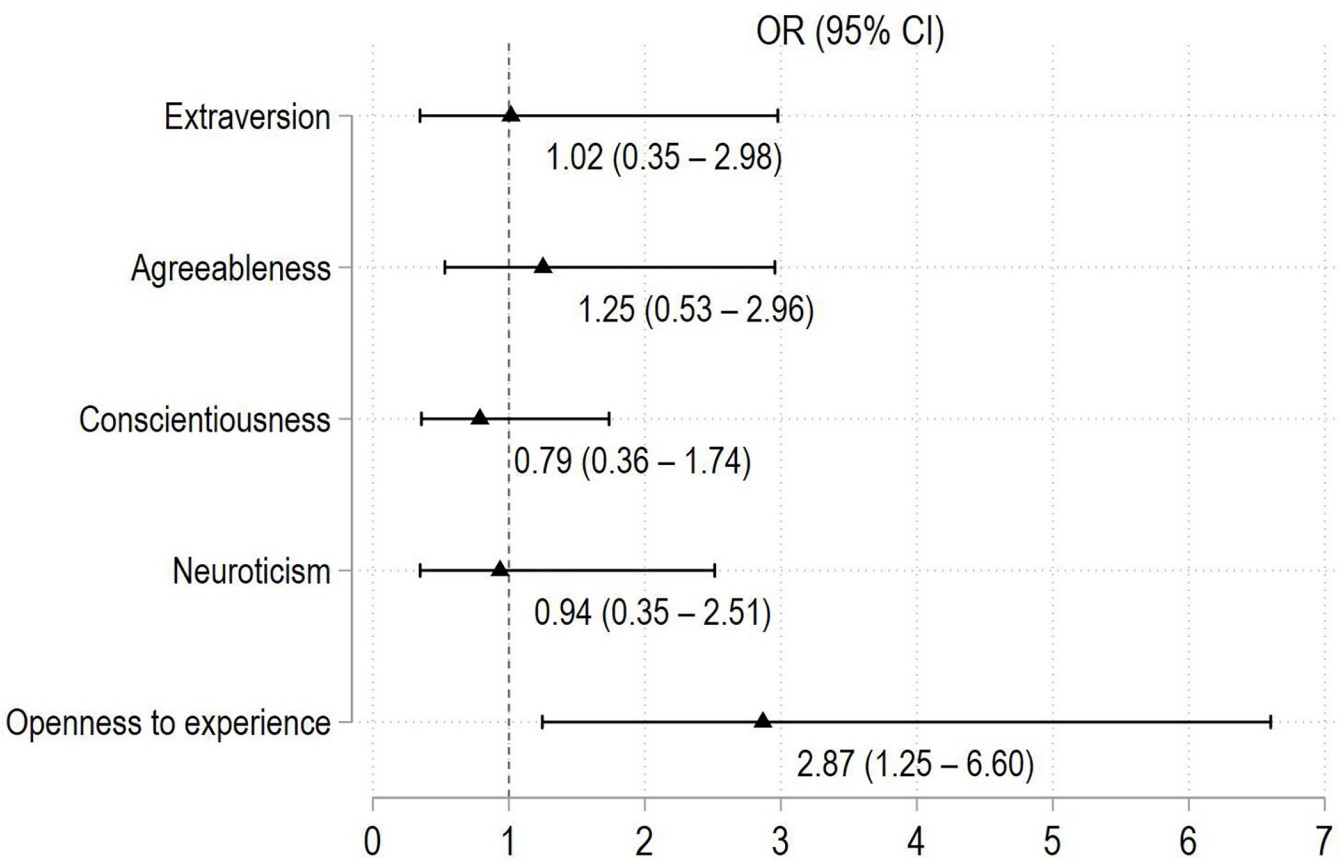

**Fig 2. Forest plot with unadjusted odds ratios for personality scores in patients with medicinal cannabis use.** Data are presented as unadjusted odds ratios with lower and upper 95% confidence intervals.

patients reported simultaneous use with their rheumatologic treatment, and few tapered their medication.

Regarding personality, we observed that neuroticism had the overall highest score while the conscientiousness trait had the lowest. Personality is a relevant construct for health-related outcomes. Bucourt et al. support this finding of high levels in neuroticism and low levels in conscientiousness traits among patients with rheumatic diseases, including RA, and further associate these traits with chronic pain [42]. When compared to the general Mexican population, our sample's average scores were lower in extraversion, agreeableness, conscientiousness, and openness to experience traits, but higher in neuroticism [26]. Personality traits can be modified by the cumulative burden associated with chronic disease trajectory, including chronic arthritis, with comorbid conditions having an additive effect on personality trait modulation [43]. There is evidence from longitudinal analyses that traits might change with chronic illness [14].

In our population, openness to experience was the only trait that showed statistical difference either in the univariate analysis or in regression models. This very broad personality domain involves a propensity to "cognitively explore", have divergent thinking, a tendency to be intellectually curious, grasp new ideas, be perceptive, imaginative, liberal, non-conforming, and actively seek out experience [13]. It has also been associated with cannabis in different clinical contexts [21]. Of note, this study focused on the conduct of MCU-seeking rather than on the proven therapeutic properties of the preparations. Nonetheless, a very high proportion

**Table 3. Multivariable logistic regression models for medicinal cannabis use on rheumatoid arthritis patients.**

| Variable | Model A | Model B | Model C |
|---|---|---|---|
| | aOR (95%CI) | | |
| Tobacco use (ASSIST moderate risk) | 3.82 (1.13–12.9) | NA | 3.36 (1.04–10.79) |
| RAPID-3 score | 1.10 (1.01–1.20) | NA | 1.10 (1.01–1.19) |
| QoL-RA score | 1.25 (0.87–1.79) | 0.97 (0.74–1.26) | 1.25 (0.91–1.72) |
| PHQ-9 score | 1.01 (0.91–1.12) | 1.01 (0.91–1.12) | NA |
| GAD-7 score | 1.07 (0.94–1.23) | 1.05 (0.92–1.19) | 1.07 (0.95–1.20) |
| Current psychiatric diagnosis | 1.72 (0.58–5.11) | 1.86 (0.65–5.28) | 1.69 (0.59–4.80) |
| Extraversion trait score | 1.12 (0.34–3.66) | 0.90 (0.29–2.76) | NA |
| Agreeableness trait score | 1.47 (0.57–3.77) | 1.19 (0.49–2.91) | NA |
| Conscientiousness trait score | 0.77 (0.31–1.87) | 0.67 (0.28–1.58) | NA |
| Neuroticism trait score | 1.63 (0.54–4.88) | 1.24 (0.44–3.47) | NA |
| Openness trait score | 2.46 (0.95–6.35) | 2.85 (1.14–7.09) | 2.81 (1.11–7.10) |
| Videoconference interview | 0.47 (0.22–1.02) | 0.53 (0.25–1.12) | 0.48 (0.22–1.02) |
| **Postestimation statistics** | | | |
| *AIC* | 217.3 | 224.1 | 209.5 |
| *Pseudo $R^2$* | 0.12 | 0.07 | 0.11 |
| *LR $X^2$ (P> $X^2$)* | 26.89 (0.008) | 16.09 (0.09) | 24.66 (0.0009) |
| *Hosmer-Lemeshow $X^2$ (P value)* | 7.67 (0.46) | 7.88 (0.44) | 5.98 (0.64) |
| *C-statistic (AUC)* | 0.72 (0.63–0.80) | 0.68 (0.58–0.77) | 0.71 (0.62–0.79) |

MCU = Medicinal cannabis use. ASSIST = Alcohol, Smoking and Substance Involvement Screening Test, BFI: Big Five inventory, RAPID-3 = Routine Assessment of Patient Index Score-3, QoL-RA = Quality of Life in RA scale, GAD-7 = Generalized Anxiety Disorder. NA = not applicable (not included in that model). aOR = adjusted odds ratios, AIC = Akaike's information criterion. LR = Likelihood ratio, AUC = Area under the curve.

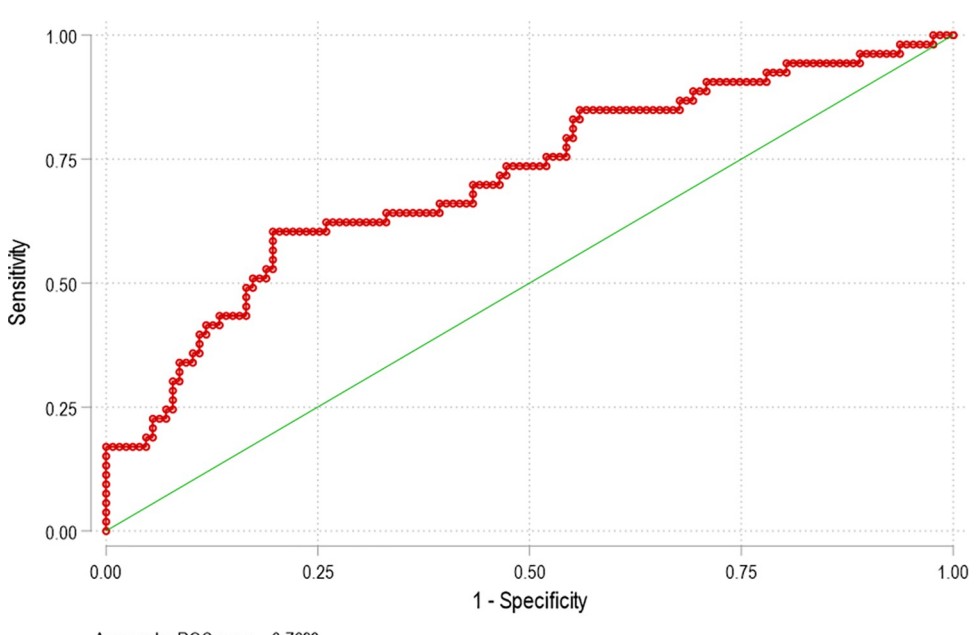

Fig 3. **Area under the curve for Model C.**

of patients reported at least some beneficial effect. There is some evidence of the openness trait as predictive of placebo response [44]. A personality-modulated use would be plausible based on a conduct-seeking pattern (openness) and a perceived efficacy for pain relief (neuroticism/ conscientiousness), even with non-standardized formulations; however, this warrants further research. Health risk perceptions might also shape a normalization experience in cannabis use [45], so the observed route of administration (mainly topical) might be playing an important role and could be culturally driven. Caballero-Hernández et al. recently described that topical herbal CAM modalities, which included formulations with cannabis, were the most widely used in patients with rheumatic diseases in a north Mexican state [40]. Concurrently, cannabis' legal status in Mexico had not yet been defined during the recruitment period of this study, so its use was informal and unregulated. As far as personality's regard in this matter, open individuals may be better able to tolerate ambiguity [13] but still prefer not to openly disclose its use. However, no difference in personality scores was observed in MCU disclosure (**Table 1 in S1 Appendix**), but we did not plan to examine this latter relationship and could not expand on this observation. This lack of disclosure was also not related to physician satisfaction or interview modality, but it has been previously described in rheumatic diseases [39], being almost universal and at a similar rate as observed in CAM [46].

Additional factors for MCU included RA disease activity/severity and tobacco use. The pain subscale was highly correlated with disease activity/severity total scores and it seemed to be a key driver in MCU. A high proportion of our patients disclosed MCU for pain crisis, hence, pain relief was one lead motivator. Moreover, previous publications highlight that chronic pain alone is the most common motivation for MCU among adults [4, 7, 36, 39]. A recent review on the perceptions of people with chronic pain who used therapeutic cannabis found positive reported effects on symptomatic alleviation, as well as in other secondary outcomes like psychological well-being [37]. When evaluated alongside personality, a higher disease activity (more pain and disability) interrelated dynamically with the openness trait, which may serve as a conductor to accept MCU. The ongoing pandemic may have contributed to this finding since there is evidence that the perception of intensity and duration of pain was related to the openness trait in chronic pain patients during the COVID-19 surge [47]. Additionally, even though openness to experience has been associated with anxiety and depression [48], no influence of these mood symptoms or current psychiatric diagnosis was observed regarding MCU or disease activity/severity in the multivariable analysis. We observed that openness scores were more associated with RA disease-related outcomes than with mental illness. Bouso et al. also observed a non-relation of self-medicated cannabis use with psychopathology in a small cohort of patients [49]. Finally, more frequent tobacco use was also associated with MCU. The existing literature is largely mixed, but a co-use has been described [50]. Lucas et al., observed an opposite pattern in which arthritis patients decreased tobacco use following oral or inhaled medical cannabis initiation [51]. We explored personality's direct relation with substance use and the openness trait score was significantly higher in moderate risk tobacco use. This was not observed on any other trait for tobacco or alcohol use (**Table 2 in S1 Appendix**). Zvolensky et al. support this observation and found a relation of the openness trait with lifetime cigarette use [52]. A complex interplay might be occurring between personality, MCU, and tobacco use motivations. It should be recognized that no safe dose for tobacco use has been established and it should be proactively assessed due to its serious health implications, particularly in the clinical context of RA and drug interaction.

Some limitations of this study need to be addressed. First, we focused on recent strictly-medicinal cannabis use, while previous studies have shown that having a familiarity with recreational cannabis use might explain MCU in patients with rheumatic diseases [38]. Second, we used a self-report measure for MCU that has not been formally validated in the Mexican

population. Although a face-to-face interview with a psychiatrist may have aided to reduce socially desirable responses and help MCU communication, self-report is prone to recall bias. Third, a current psychiatric diagnosis was established through the health records. A structured evaluation could be employed in further research. Fourth, the study took place during the COVID-19 pandemic and this may have influenced the results in terms of RA-related outcomes, psychiatric comorbid conditions, and MCU. Fifth, the cross-sectional nature of the design cannot establish causal relationships. Sixth, we did not use specific tools to assess disease activity, such as the Clinical Disease Activity Index (CDAI), the Simplified Disease Activity Index (SDAI), or the Disease Activity Score-28 joints evaluated (DAS28), due to the limited possibility of in-person-consultations required for joint counts. Instead, we use the RAPID-3, a self-administered questionnaire that provides similar information to the DAS28 and CDAI [28] in different populations, including patients from the Latin-American region [53]. Last, the perception of the impact of physical symptoms depends on many factors that include the individual's coping style, emotion regulation, medication adherence, and subjective well-being. These associations were not explored here and should be considered in future investigations.

## Conclusions

The present study contributes to the perspective of MCU in RA patients and shows that its use is frequent and explained by individual factors that include personality. The route in the administration of MCU might be relevant and should be kept in mind as it may reveal a specific scenario in cannabis use. Personality assessment is important under the biopsychosocial approach. A comprehensive underpinning of the patient's inner perspective, motivations, and conduct is necessary to adopt a patient-centered approach in the everyday clinical scenario. Additional factors include rheumatic disease activity/severity and moderate tobacco use and these aspects need to be thoroughly addressed with patients. On the contrary, sociodemographic factors, disease duration, comorbidity, health care satisfaction, and psychopathology did not play an important role for MCU, although they are important determinants of health. A proactive and direct clinical assessment of MCU from an interdisciplinary perspective is relevant as it may uncover important health information to help adjust treatment goals and enhance related outcomes.

## Supporting information

**S1 Appendix. Supplementary material.** Supplementary Data.
(DOCX)

## Acknowledgments

We thank all included participants as well as social service interns who contributed in patient recruitment.

## Author Contributions

**Conceptualization:** José R. Galindo-Donaire, Gabriela Hernández-Molina, Ana Fresán Orellana, Oswaldo Briseño-González, Virginia Pascual-Ramos.

**Data curation:** José R. Galindo-Donaire, Irazú Contreras-Yáñez.

**Formal analysis:** José R. Galindo-Donaire, Ana Fresán Orellana, Irazú Contreras-Yáñez.

**Investigation:** José R. Galindo-Donaire, Guillermo Guaracha-Basáñez, Virginia Pascual-Ramos.

**Methodology:** José R. Galindo-Donaire, Gabriela Hernández-Molina, Ana Fresán Orellana, Irazú Contreras-Yáñez, Guillermo Guaracha-Basáñez, Oswaldo Briseño-González, Virginia Pascual-Ramos.

**Project administration:** José R. Galindo-Donaire.

**Supervision:** José R. Galindo-Donaire, Gabriela Hernández-Molina, Ana Fresán Orellana, Virginia Pascual-Ramos.

**Validation:** Gabriela Hernández-Molina, Ana Fresán Orellana, Irazú Contreras-Yáñez, Guillermo Guaracha-Basáñez, Virginia Pascual-Ramos.

**Visualization:** José R. Galindo-Donaire.

**Writing – original draft:** José R. Galindo-Donaire, Virginia Pascual-Ramos.

**Writing – review & editing:** José R. Galindo-Donaire, Gabriela Hernández-Molina, Ana Fresán Orellana, Irazú Contreras-Yáñez, Guillermo Guaracha-Basáñez, Oswaldo Briseño-González, Virginia Pascual-Ramos.

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
