## [Decision Letter · Decision Letter 0]

3 Nov 2022

PONE-D-22-14277The role of personality traits on self-medicated cannabis in rheumatoid arthritis patients: A multivariable analysisPLOS ONE

Dear Dr. Pascual-Ramos,

Thank you for submitting your manuscript to PLOS ONE. After careful consideration, we feel that it has merit but does not fully meet PLOS ONE’s publication criteria as it currently stands. Therefore, we invite you to submit a revised version of the manuscript that addresses the points raised during the review process.

We look forward to receiving your revised manuscript.

Kind regards,

Christine Nardini

Academic Editor

PLOS ONE

Journal Requirements:

Reviewers' comments:

Reviewer's Responses to Questions

**Comments to the Author**

1. Is the manuscript technically sound, and do the data support the conclusions?

Reviewer #1: Yes

Reviewer #2: Partly

2. Has the statistical analysis been performed appropriately and rigorously? 

Reviewer #1: Yes

Reviewer #2: I Don't Know

3. Have the authors made all data underlying the findings in their manuscript fully available?

Reviewer #1: No

Reviewer #2: Yes

4. Is the manuscript presented in an intelligible fashion and written in standard English?

Reviewer #1: Yes

Reviewer #2: Yes

5. Review Comments to the Author

Reviewer #1: The authors investigated the role of personality traits on medical use of cannabis in RA patients. I feel that the paper contains information of interest to readers. however, the revisions/modifications listed below will, in my opinion, help empower the manuscript even more.

Background section

Line 54: the sentence “lack of a known cure using traditional medicine” is imprecise. To date, due to the introduction of effective and expensive biological and synthetic Disease Modifying Anti-Inflammatory Drugs (DMARDs), sustained remission status is more likely achievable compared to the past, although residual synovitis can support the progression of bone damage occurring in some patients.

Line 56: the reference to the cited study should be more accurate. Bruce D et al. reports that patients with certain chronic conditions (also RA?) use medical cannabis both as a complementary method for symptom management and treatment of standard-of-care medication side-effects.

Groups and outcome

The authors proposed to use a cut-off points for sample size calculation and population description of personality traits, however I not advocate making categorical variable out of continuous variables because there might be loss of information.

Describe the definition of a positive response based on the modified ICAM-Q: How the rationale for the medical use of cannabis is defined? Based on what criteria?

Statistical analysis section

Line 161: STATA has two version of AIC statistics, one used with -glm- and another -estat ic-. The -estat ic- version does not adjust the log-likelihood and penalty term by the number of observations in the model, whereas the version used in -glm- does. Having an adjustment by the number of observations is preferable.

Population characteristics

As you previously define “ever history of severe mental illness, including psychotic manifestations, or current severe psychiatric diagnosis that hindered interview performance or completetion” (which correspond to patients scored CGI-S>=6) a criteria of exclusion, the “current psychiatric diagnosis of one in ten patients” (line 182) should be clearly described by the relative CGI-S score.

MCU and related information section

Line 201 and following: please revise all the number and percentage. A total of 53 patients reported MCU, which not correspond to the total of patients categorized based on the type of MCU (topical, oral or smoked).

Multivariable analysis section

The details of the 3 models with the p values of each independent variable should be entered in Table 3. Indeed, in line 237 you describe a statistical trend of the openness trait to be independently associated to MCU in model A (aOR=2.46, 0.95-6.35, p=0.06), but this value is not given in table 3.

In addition, logistic regression does not have an equivalent means to the R squared that is found in OLS regression (i.e. the proportion of variance explained by the predictors). So, I suggest interpreting the pseudo R squared statistics with great caution. In addition, the pseudo R2 values given, are very low so it would be useful to know the prob>chi2 of each model, which defines the probability that the null hypothesis is true, meaning that there is no effect of the independent variables taken together, on the dependent variable.

I also suggest that the authors describe the regression model-based analysis, if stepwise or backward-selection. The backward instead of stepwise automated variable selection is preferable, with the identification of the variables that are not important in the multivariate model through the Would test, the evaluation of significance of the excluded and retained variables through the partial likelihood ratio test, the definition of the correct parametric form for the proper variables and the elaboration of a list of possible interactions between variables.

In addition, the authors should define the quality of the model by AUC calculation and relative graph inclusion, which helps provide a visual interpretation.

Discussion section

Line 282: the mentioned analysis both report a p value of 0.09 that cannot be considered a trend of statistical significance, even more so because of the lack of statistical power.

Reviewer #2: The paper is well designed and interesting, and it will give further informations to clinicians about something concerning self medicated cannabis use in rheumatoid arthritis. Nevertheless additional factors for medical cannabis use are reported to be disease activity/disease severity but in your paper no data about disease activity such as DAS28 (Prevoo ML, van ‘t Hof MA, Kuper HH, van Leeuwen MA, van de Putte LB, van Riel PL. Modified disease activity scores that include twenty-eight-joint counts. Development and validation in a prospective longitudinal study of patients with rheumatoid arthritis. Arthritis Rheum. 1995;38:44–48), CDAI or SDAI are mentioned but only questionnaires usually used as seft reported outcomes.

In order to complete your data and make the manuscript unambiguous about data concerning disease activity I suggest to add these informations and repeat statistical analyses adding these clinical indices

6. PLOS authors have the option to publish the peer review history of their article (what does this mean?). If published, this will include your full peer review and any attached files.

Reviewer #1: No

Reviewer #2: No

---

## [Author Response · Author response to Decision Letter 0]

30 Nov 2022

RESPONSES TO REVIEWERS

Reviewer #1: The authors investigated the role of personality traits on medical use of cannabis in RA patients. I feel that the paper contains information of interest to readers. however, the revisions/modifications listed below will, in my opinion, help empower the manuscript even more.

Response: We appreciate the reviewer general comment.

Background section

Line 54: the sentence “lack of a known cure using traditional medicine” is imprecise. To date, due to the introduction of effective and expensive biological and synthetic Disease Modifying Anti-Inflammatory Drugs (DMARDs), sustained remission status is more likely achievable compared to the past, although residual synovitis can support the progression of bone damage occurring in some patients.

Response: We agree with the reviewer and propose the update sentence that includes a recent publication from our group.

“Some cited reasons for alternative use have been the chronic and painful nature of the disease, adverse events associated with DMARDs use, and the scientific perception that the disease lacks a known cure [3-5], in spite of recent evidence that sustained drug-free remission, a proxy for a cure, can be achieved in particular patients [6].”

Line 56: the reference to the cited study should be more accurate. Bruce D et al. reports that patients with certain chronic conditions (also RA?) use medical cannabis both as a complementary method for symptom management and treatment of standard-of-care medication side-effects.

Response: We have clarified that Bruce’s reference also includes RA patients. 

Groups and outcome

The authors proposed to use a cut-off points for sample size calculation and population description of personality traits, however I not advocate making categorical variable out of continuous variables because there might be loss of information.

Response: We agree with the reviewer. However, to achieve the primary objective (association between personality traits and MCU), personality traits were included in the analyses as continuous variables, which does not invalidate our results.

Regarding sample size calculation, it will be questionable to change what we did to define the number of patients needed to accomplish the primary objective. We have repeated sample size calculation considering personality traits as a continuous variable, and information has been added to the corresponding section. We hope this proposal will satisfy the reviewer. 

Describe the definition of a positive response based on the modified ICAM-Q: How the rationale for the medical use of cannabis is defined? Based on what criteria?

Response: We have provided a brief description of ICAM-Q, the modification of the instrument and how MCU was defined. The questionnaire includes a section where patients are directed to identify the main reason for MCU and to evaluate how helpful they found the product. We have added the information in the “Groups and Definitions” section.

Statistical analysis section

Line 161: STATA has two version of AIC statistics, one used with -glm- and another -estat ic-. The -estat ic- version does not adjust the log-likelihood and penalty term by the number of observations in the model, whereas the version used in -glm- does. Having an adjustment by the number of observations is preferable.

Response: We thank the reviewer for the comment. We would like to point out that the number of observations (n=180) remained stable throughout all models, thus the numerator for adjusting log-likelihood and penalty term would not change in the calculations. Using glm is useful when comparing models with different sample sizes. Nonetheless, we ran the analysis using glm model for calculating AIC and found that values of AIC were still lowest for model C, serving the intended purpose for final model selection (avoiding underfitting/overfitting).

Values of AIC using glm:

Madel A: 1.255845

Model B: 1.297252

Model C: 1.211032

Population characteristics

As you previously define “ever history of severe mental illness, including psychotic manifestations, or current severe psychiatric diagnosis that hindered interview performance or competition” (which correspond to patients scored CGI-S>=6) a criteria of exclusion, the “current psychiatric diagnosis of one in ten patients” (line 182) should be clearly described by the relative CGI-S score.

Response: We have added that the totality of the patients with current psychiatric conditions were below the fifth category threshold (CGI-S score≤5). 

MCU and related information section

Line 201 and following: please revise all the number and percentage. A total of 53 patients reported MCU, which not correspond to the total of patients categorized based on the type of MCU (topical, oral or smoked).

Response: We have clarified, in the corresponding section, that concurrent use of different administration routes (oral, topical and smoked) was possible and were not mutually exclusive; so percentages do not necessarily add up to 100%. 

Multivariable analysis section

The details of the 3 models with the p values of each independent variable should be entered in Table 3. Indeed, in line 237 you describe a statistical trend of the openness trait to be independently associated to MCU in model A (aOR=2.46, 0.95-6.35, p=0.06), but this value is not given in table 3.

Response: We have adopted the suggestion and included all the data required in table 3. 

In addition, logistic regression does not have an equivalent means to the R squared that is found in OLS regression (i.e. the proportion of variance explained by the predictors). So, 

I suggest interpreting the pseudo-R squared statistics with great caution. In addition, the pseudo R2 values given, are very low so it would be useful to know the prob>chi2 of each model, which defines the probability that the null hypothesis is true, meaning that there is no effect of the independent variables taken together, on the dependent variable.

Response: We agree with the reviewer. Proportion of variance cannot be explained by Pseudo R2 in logistic regression, at least not by McFadden’s calculation. Presenting Pseudo R2 in addition to other postestimation statistics helped us select the best fitted regression model, that is, comparing Pseudo R2, AIC, Hosmer-Lemeshow test and C-statistic among models. On the other hand, although we did not achieve a McFadden’s Pseudo R2 between 0.2-0.4 that, according to some authors, would indicate a “good” model, interpretation alongside the other statistics helped us provide information that our selected model was the best with the available data, while avoiding underfitting or overfitting.

We have included LR of Chi2 and probability >chi2 in table 3.

I also suggest that the authors describe the regression model-based analysis, if stepwise or backward-selection. The backward instead of stepwise automated variable selection is preferable, with the identification of the variables that are not important in the multivariate model through the Would test, the evaluation of significance of the excluded and retained variables through the partial likelihood ratio test, the definition of the correct parametric form for the proper variables and the elaboration of a list of possible interactions between variables.

Response: We have clarified, in the corresponding section, that a test-based backward selection was used for analysis. Wald’s test and p-value were the criteria for exclusion or retention of variables across models. Interactions of variables with personality are described in the model and presented in the results. 

In addition, the authors should define the quality of the model by AUC calculation and relative graph inclusion, which helps provide a visual interpretation.

Response: C-statistic is equal to the AUC and is reported in table 3 as a postestimation statistic. We have adopted the suggestion and included a graph for visual interpretation.

Discussion section

Line 282: the mentioned analysis both report a p value of 0.09 that cannot be considered a trend of statistical significance, even more so because of the lack of statistical power.

Response: We have deleted the sentence.

Reviewer #2: The paper is well designed and interesting, and it will give further informations to clinicians about something concerning self medicated cannabis use in rheumatoid arthritis. Nevertheless additional factors for medical cannabis use are reported to be disease activity/disease severity but in your paper no data about disease activity such as DAS28 (Prevoo ML, van ‘t Hof MA, Kuper HH, van Leeuwen MA, van de Putte LB, van Riel PL. Modified disease activity scores that include twenty-eight-joint counts. Development and validation in a prospective longitudinal study of patients with rheumatoid arthritis. Arthritis Rheum. 1995;38:44–48), CDAI or SDAI are mentioned but only questionnaires usually used as self reported outcomes.

In order to complete your data and make the manuscript unambiguous about data concerning disease activity I suggest to add these informations and repeat statistical analyses adding these clinical indices.

Response: We appreciate the reviewer's comment. The study was developed during the current pandemic (June 2020-August 2021). In March 2020, the Mexican government declared our Institution a dedicated COVID-19 hospital, and visits to the Department of Immunology and Rheumatology outpatient clinic were interrupted and moved to phone medical consultations. Nonetheless, given the middle-low socioeconomic status of most of our patients and the limited resources available at our Institution, the move was challenging. In June 2020 (when the current study was initiated), the outpatient clinic was progressively reinstalled, while patients at risk for severe manifestations (those with Systemic Lupus Erythematosus, Systemic Vasculitis, Systemic Sclerosis, and IgG4 Disease) were prioritized for in-person consultations. Today, only 60% of patients currently receive face-to-face consultations. In such a context, a physical examination, which is required to score a DAS28 (CDAI and SDAI), was unrealistic for the potential candidates for the current study, with the additional concern of a risk of bias (RA patients at risk for more severe manifestations would be offered in-person visits). With the above considerations in mind, we opted for the RAPID-3. In usual clinical care, the clinician makes decisions based not only on an index but on a composite of findings from the patient history, the physical examination, laboratory tests, and imaging data. RAPID3 seems sufficient to document patient status quantitatively, to compare from one visit to another, and to document a course over long periods (Pincus et al. RAPID3, an Index to Assess and Monitor Patients with Rheumatoid Arthritis, Without Formal Joint Counts: Similar Results to DAS28 and CDAI in Clinical Trials and Clinical Care. Rheum Dis Clin N Am 35 (2009) 773–778). RAPID-3 also has the advantage of being tested in RA Latin American patients (Ballesteros-Muñoz G et al. Correlation between rapid-3, DAS28, CDAI and SDAI as a measure of disease activity in a cohort of Colombian patients with rheumatoid arthritis. Clin Rheumatol. 2017 May;36(5):1143-1148). 

However, we agree with the reviewer. RAPID3 does not replace indices that include joint counts and laboratory tests, and both, the Core Data Set and DAS28 are preferred for their specificity. We have added this as a limitation.

---

## [Decision Letter · Decision Letter 1]

23 Dec 2022

The role of personality traits on self-medicated cannabis in rheumatoid arthritis patients: A multivariable analysis

PONE-D-22-14277R1

Dear Dr. Pascual-Ramos,

We’re pleased to inform you that your manuscript has been judged scientifically suitable for publication and will be formally accepted for publication once it meets all outstanding technical requirements.

Kind regards,

Christine Nardini

Academic Editor

PLOS ONE

Additional Editor Comments (optional):

Reviewers' comments:

Reviewer's Responses to Questions

**Comments to the Author**

1. If the authors have adequately addressed your comments raised in a previous round of review and you feel that this manuscript is now acceptable for publication, you may indicate that here to bypass the “Comments to the Author” section, enter your conflict of interest statement in the “Confidential to Editor” section, and submit your "Accept" recommendation.

Reviewer #1: All comments have been addressed

Reviewer #2: All comments have been addressed

2. Is the manuscript technically sound, and do the data support the conclusions?

Reviewer #1: Yes

Reviewer #2: (No Response)

3. Has the statistical analysis been performed appropriately and rigorously? 

Reviewer #1: Yes

Reviewer #2: (No Response)

4. Have the authors made all data underlying the findings in their manuscript fully available?

Reviewer #1: Yes

Reviewer #2: (No Response)

5. Is the manuscript presented in an intelligible fashion and written in standard English?

Reviewer #1: Yes

Reviewer #2: (No Response)

6. Review Comments to the Author

Reviewer #1: The authors have addressed my concerns and made the manuscript acceptable for publication. I thank the authors for their attention in making the requested changes.

Reviewer #2: (No Response)

7. PLOS authors have the option to publish the peer review history of their article (what does this mean?). If published, this will include your full peer review and any attached files.

Reviewer #1: No

Reviewer #2: No

---

## [Editor Report · Acceptance letter]

3 Jan 2023

PONE-D-22-14277R1 

The role of personality traits on self-medicated cannabis in rheumatoid arthritis patients: A multivariable analysis 

Dear Dr. Pascual-Ramos:

I'm pleased to inform you that your manuscript has been deemed suitable for publication in PLOS ONE. Congratulations! Your manuscript is now with our production department. 

Kind regards, 

on behalf of

Dr. Christine Nardini 

Academic Editor

PLOS ONE